# Autoimmunity, New Potential Biomarkers and the Thyroid Gland—The Perspective of Hashimoto’s Thyroiditis and Its Treatment

**DOI:** 10.3390/ijms25094703

**Published:** 2024-04-26

**Authors:** Ewa Tywanek, Agata Michalak, Joanna Świrska, Agnieszka Zwolak

**Affiliations:** 1Department of Internal Medicine and Internal Medicine in Nursing, Medical University of Lublin, Witold Chodźki Street 7, 20-093 Lublin, Poland; ewa.tywanek@umlub.pl (E.T.); agnieszka.zwolak@umlub.pl (A.Z.); 2Doctoral School, Medical University of Lublin, 20-093 Lublin, Poland; 3Endocrinology Department with Nuclear Medicine Department, Center of Oncology of the Lublin Region St. Jana z Dukli, Kazimierz Jaczewski Street 7, 20-090 Lublin, Poland; joannaswirska@wp.pl; 4Department of Gastroenterology, Medical University of Lublin, Poland, Jaczewski Street 8, 20-954 Lublin, Poland

**Keywords:** thyroid gland, thyroiditis, autoimmunity, antibodies, biomarkers

## Abstract

Autoimmune thyroid disease (AITD) is the most common organic specific illness of the thyroid gland. It may manifest as the overproduction or the decline of thyroxine and triiodothyronine. Hyperthyroidism develops due to the overproduction of hormones as an answer to the presence of stimulatory antibodies against the TSH receptor. Hashimoto’s thyroiditis (HT) is generally characterized by the presence of thyroid peroxidase and thyroglobulin antibodies, with a concomitant infiltration of lymphocytes in the thyroid. Due to the progressive destruction of cells, AITD can lead to subclinical or overt hypothyroidism. Pathophysiology of AITD is extremely complicated and still not fully understood, with genetic, environmental and epigenetic factors involved in its development. Due to increasing incidence and social awareness of this pathology, there is an urgent need to expand the background concerning AITD. A growing body of evidence suggests possible ways of treatment apart from traditional approaches. Simultaneously, the role of potential new biomarkers in the diagnosis and monitoring of AITD has been highlighted recently, too. Therefore, we decided to review therapeutic trends in the course of AITD based on its pathophysiological mechanisms, mainly focusing on HT. Another aim was to summarize the state of knowledge regarding the role of new biomarkers in this condition.

## 1. Introduction

Thyroiditis constitutes a medical condition of special concern due to its diversified background. Autoimmune factors, inflammatory reasons, drug-induced occurrences or fibrotic process might constitute the background of its development. Among these potential scenarios, the most common one constitutes autoimmune thyroiditis (Hashimoto’s thyroiditis (HT), Graves’ Disease, postpartum thyroiditis or painless sporadic thyroiditis). HT, also known as chronic lymphocytic or autoimmune thyroiditis, is the most common, an organ-specific autoimmune disease in iodine-sufficient areas, affecting the thyroid gland. In some cases, this pathology may lead to the development of hypothyroidism [1,2]. Additionally, a viral infection, with the viral structure being similar to the thyroid protein, is believed to induce the synthesis of antibodies against the thyroid gland. HT is characterized by the presence of specific autoantibodies, such as thyroid peroxidase (TPOAb) and thyroglobulin (TgAb) antibodies. Both HT and Graves’ Disease are classified as autoimmune thyroid disorders (AITDs), sometimes called “the opposite poles of the same disease” [3,4]. Their natural history is modified by various influencing factors, e.g., infection, genetic and environmental reasons. Therefore, a clinical manifestation might be significantly diversified Autoimmune thyroiditis has been originally discovered by Hakaru Hashimoto in 1912 (1). He identified thyroidectomized gland as an enlarged organ infiltrated with lymphocytes. The connection between this finding and the presence of autoantibodies was unknown till 1956, when Roitt, Doniach et al. discovered antithyroglobulin antibodies in the serum of patient with lymphocytic goiter, suggesting this state to be an autoimmune pathology [5,6]. HT is not only the most common autoimmune endocrine disease [2,7], but it is also the major cause of hypothyroidism in iodine-sufficient areas [8]. The overall prevalence of HT is about 7.5%, with a prevalence of 17.5% in women and 6.0% in men, depending on the geographic region [9]. As mentioned above, predominance of illness in females is clearly noticeable, as only about 10% of affected individuals are men [10]. It seems that a growing overall incidence of hypothyroidism [11] and autoimmune thyroiditis [12] can be triggered by greater exposure to iodine. Autoimmune thyroiditis leads to the chronic inflammation of the thyroid gland tissue, and might be followed by hypothyroidism in about 25% of patients [13]. Deficiency in thyroid hormones may be subclinical or overt, depending on free thyroxine (fT4) and free triiodothyronine (fT3) levels and the clinical manifestation of the disease. A typical combination of symptoms (including fatigue, cold intolerance, weight gain) is the result of generalized reduction in metabolic processes and/or the accumulation of matrix glycosaminoglycans in the interstitial spaces (coarse hair or hoarseness of voice) [14]. The final intensity of complaints is varied. In children, hypothyroidism may manifest with goiter, delayed puberty or growth retardation [14]. Hypothyroidism due to HT becomes more common in advanced age, with a peak of onset between 40 to 60 years [8]. Substantial differences in geographic variability in the prevalence and incidence concerning HT are also observed among patients [3]. The coexistence of AITD with different autoimmune disorders is another commonly observed dependence [15] (Figure 1).

## 2. Chronic Autoimmune Thyroiditis—A General Overview

Genetic and environmental factors are known to be involved in the pathogenesis of chronic autoimmune thyroiditis [3], whereas epigenetics factors [16] are also indicated as important players in the development of the disease. All of the above-mentioned factors cause a dysregulation of the immune system, followed by abnormal function of the innate and humoral response. Chronic autoimmune thyroiditis has been associated with the failure of T cell-mediated inflammatory pathways through complex mechanisms involving antigen presenting T cells and B cells. Infiltration of thyroid tissue with lymphocytes, mainly T helper 1 (Th1) cells, can directly alter thyroid gland function through the mediation of interleukin-1 (IL-1), tumor necrosis factor (TNF) and interferon gamma (IFN-γ) [14]. In some cases, infiltration of the thyroid with lymphocytes leads to the deposition of fibrotic tissue in the thyroid and subsequently causes hypothyroidism. The infiltration of the thyroid with lymphocytes may be observed as a hypoechogenic pattern in ultrasonography [17]. What is more, ultrasonography mode can be supported by artificial intelligence. This combination might constitute a potential breakthrough in the accurate diagnosis of the classic form of HT, as well as an antibody-negative chronic autoimmune thyroiditis (SN-CAT) [18]. The assessment of the serum concentration of TPOAb and TgAb, which are serum biomarkers of HT, plays a fundamental role not only in the clarification of a diagnostic approach in the course of illness, but also in the prediction of progression regarding thyroid-followed immunity to hypothyroidism [19]. Autoantibodies directed against different minor thyroid antigens, such as pendrin, sodium iodide symporter and megalin, are also known. However, they are not widely used is the clinical practice due to the lack of sufficient data describing their potential function [20]. Based on its etiology, HT may be classified as primary (with no known direct cause of the illness) or secondary (due to identifiable factor) forms [2]. Histologically, several main types of a primary HT may be distinguished: a classic form, fibrous variant, IgG4-related subtype, juvenile form, Hashitoxicosis and, finally, sporadically developed postpartum painless or silent thyroiditis [2]. Nevertheless, regardless of the certain background of primary HT, the common feature of pathological findings in the thyroid gland is its lymphocytic infiltration. Moving further, the secondary form of HT is the iatrogenic one in the majority of cases, with a known etiological cause, e.g., due to immunomodulatory therapy—for example interferon-alpha (IFN-α) or immunotherapy for cancer [2]. As the etiology of chronic thyroiditis is multifactorial, a potential causal treatment seems to be a complicated solution. Thus, the main broadly accepted type of management is L-thyroxine substitution therapy. Considering the still increasing prevalence of the illness and its significant impact on quality of life, new biomarkers assessing different aspects of immune system dysfunction and its pathogenesis are urgently required to be investigated. Alternative or complementary types of therapeutic procedures should be continuously researched and implemented towards incomplete response to basic treatment in all of the individuals with HT. The interest of researchers should be directed into the development of new serological parameters useful in the monitoring of the disease, as well as verifying the efficacy of ongoing treatment. Nowadays, the already known and novel markers of AITD are considered to be used in everyday clinical practice (Figure 2). Nonetheless, surveys in this area are still conducted and some new molecular pathways worth exploring them are still in front of us. Fetal cell microchimerism (FCM) and maternal cell microchimerism (MCM) belong to them. They constitute phenomena that take place during pregnancy, when the transfer of cells, including some from the immune systems, can occur either from the fetus to the mother (FCM) or from the mother to the fetus (MCM). Thus, the increased risk of AITD might be explained in several ways. FCM induces a graft-versus-host reaction (due to infectious or environmental factors, drugs, or abnormal tissue proteins), and it breaks the maternal tolerance to fetal cells and results in AITD. Subsequently, the molecular mimicry mechanism between gut and thyroid tissue transglutaminase might be the reason for thyroiditis in patients with celiac disease. Predisposition to AITD may be also explained with some kind of previously underwent viral infections. Molecular mimicry together with the presence of viral and bacterial superantigens are the examples of such causative factors. From the genetic background, FOX3P3 gene is known as a key player in the production of T regulatory cells. Some surveys have shown to present a notable relationship between them and the development of AITD. HLA-DR3 constitutes another molecular particle more common among patients with autoimmune thyroiditis [2].

## 3. Oxidative Stress in Thyroid Disorders

Oxidation–reduction (redox) homeostasis is essential to maintain proper functions of vital processes. Reactive oxygen species (ROS) belong to subcellular messengers in signal transductions pathways [21]. ROS are primarily generated by mitochondria [22] or produced as byproducts in reactions conducted via specific enzymes, such as glutathione peroxidase, superoxide dismutases, peroxiredoxins, myeloperoxidases or catalase [21]. Oxidative stress is generally known as “a disturbance in the prooxidant–antioxidant balance in favor of the former” [23]. Reactive oxygen species are very short-lived, so the direct measurement of their concentrations appears to be almost impossible [24]. Therefore, identifiable metabolites generated due to oxidative damage of cell components (proteins, lipids and DNA) are used to measure the intensity of the redox state—these molecules are known as specific biomarkers; 8-hidroxy-2-deoxyguanosine (8-OHdG) [25], malondialdehyde (MDA) [26] or total antioxidant capacity (TAC) measurements [27] may be pointed as examples [28]. Oxidative stress (OS) is considered to be involved in the development of many various diseases, e.g., cancer [29], diabetes [30], cardiovascular [21] or liver disorders [31], Alzheimer’s disease [32], psychological disturbances [33], pathological states affecting reproductive health [34,35,36] any many others. HT is also known to be connected with OS [37], as well as in children and adolescents [27]. None of the already known biomarkers of OS are currently widely accepted as the best tool to be implied in HT, so researchers use different novel molecules within investigations. In the group of patients with subclinical hypothyroidism and HT, mean total oxidant status (TOS) and oxidative stress index (OSI) were significantly higher in individuals, who developed overt disorders, which raised the suspicion that oxidative stress may be an effective risk factor in the development of overt hypothyroidism in the course of HT [38]. According to already achieved data, mitochondrial DNA copy number (mtDNAcn) and mtDNA damage could be also perceived as other possible markers of OS in patients with HT [39,40]. It is worth mentioning here that microRNAs (miRNAs) are the small non-coding particles of RNA that regulate gene expression at the posttranscriptional level. They were reported as emerging potential biomarkers and a therapeutic target for several diseases, including AITD [41]. One of the studies was established to explore OS-related genes potentially involved in the course of HT [42]. Xu et al. managed to isolate a few of them and even to indicate notable relationships between certain genes and the expression of selected miRNAs. Interestingly, evident dependences between immune system cells (natural killer cells, B cells) and selected genes among HT patients were revealed. Such observations highlight the need to explore a genetic and immune background of thyroid disorders in the aspect of their treatment. Another exploration in this field showed advanced glycation end products (AGEs) to be potential biomarkers of OS in individuals with HT. Additionally, the expression of antioxidant paraoxonase (PON-1) was identified to be significantly decreased. Of note, AGEs and PON-1 correlated with each other conversely, emphasizing the phenomenon of inflammatory and pro-oxidant imbalance in the course of HT [43]. Focusing on the amelioration of oxidative imbalance, adjuvant therapies based on selenium, zinc and vitamins D and C can be perceived as potential additional strategies in patients with thyroiditis [44].

## 4. Possible Therapeutic Trends in Chronic Thyroiditis—What Do We Know and What Should We Improve?

### 4.1. L-Thyroxine Replacement Therapy

L-thyroxine substitution therapy is the main broadly accepted and used approach in the treatment of overt and subclinical (in particular cases) hypothyroidism due to chronic thyroiditis. The administration of L-thyroxine, which is a synthetic form of T4, should be individually adapted to each patient after the detailed assessment of the entire clinical picture. Full replacement doses in adults are approximately 1.6 ug/kg/day [8]. L-thyroxine taken orally undergoes absorption in the small bowel, and thus, patients with gastric pathologies may require higher doses of the drug. L-thyroxine requirement rises due to increasing gastric pH, as well as the progressive damage of gastric mucosa [45]. However, the majority of patients treated with L-thyroxine achieve normalization of TSH and thyroid hormones levels. Subsequent acceleration of metabolism is the direct factor responsible for quality of life improvements. T4 replacement therapy for subclinical hypothyroidism with HT results in lipid profile improvement [46], with positive changes in the relationships between protective and proatherogenic fractions of serum lipids, as well as optimizing blood pressure [47]. What is interesting, T4 replacement therapy might reverse cardiac dysfunction in patients with subclinical hypothyroidism or even in individuals with normal TSH level [48]. Apart from studies reporting the beneficial influence of L-thyroxine therapy on metabolic status among treated individuals, there are some data regarding the exact impact of the therapy on thyroid volume. In a group of 50 euthyroid non-goitrous children and adolescents with autoimmune thyroiditis, treatment with L-thyroxine was shown to significantly reduce thyroid volume [49]. According to already-performed analyses, L-thyroxine therapy may also exert a direct effect on immunological parameters. Guclu et al. performed a study assessing the influence of the therapy on chosen immunological parameters. In a group of 65 female patients with HT, after 10–12 weeks of L-thyroxine treatment and simultaneous restoring euthyroid state, not only was there a statistically significant decrease in the serum levels of TSH, TgAb and TPOAb, a concomitant increase in fT4 serum levels was achieved. A statistically notable decrease in serological concentration of IL-12 was another essential finding. Of note, the decrease in interferon gamma (IFN-γ) serum levels was not statistically significant and there were no changes in serum IL-2 and IL-4 [50]. The interpretation of above-mentioned results can be explained by the inhibition of the inflammatory process due to the suppression of T helper type 1 cells [50]. On the other hand, the results of different study have suggested the presence of the direct effect of L-thyroxine therapy on the percentages of selected populations of dendritic cells (DCs), with a significant reduction among their plasmacytoid subtype in peripheral blood and no changes concerning the total number of conventional DCs. However, the cluster of conventional DCs expressing CD86 and CD91 was shown to have a greater percentage [51]. The normalization of serum TSH level doses does not have to optimize serum T3; it concerns particularly individuals with a decreased effectiveness of conversion between T4 and T3. This occurrence might be observed among patients affected with polymorphism of iodothyronine deiodinase 2 (DIO2) [14]. Thereby, a combination therapy with both LT4 and LT3 is considered; however, there is an ongoing worldwide debate on this issue and trustworthy parameters that should be used to make adjustments in therapy, having in the mind the short half-life of T3 [8]. With regard to the above-mentioned concerns, therapy based on L-thyroxine (with or without T3), which has seemed to be only a symptomatic one, turned out to exert a simultaneous immunomodulatory effect. The additional implication of new biomarkers assessing the immune system in treated patients would constitute a perfect approach.

### 4.2. Iodine

Iodine is an absolutely necessary microelement for a proper function of the thyroid. Follicles of the gland constitute a warehouse for thyroglobulin, which is a substrate for thyroid hormones production. Iodine deficiency in childhood affects the growth adversely; in adults, it may trigger various thyroid disorders. Severe deficiency of this microelement causes goiter and hypothyroidism. In a mild-to-moderate iodine deficiency, a tendency towards a higher prevalence of toxic nodular goiter and hyperthyroidism might be observed. On the other hand, an increased iodine intake in an iodine-deficient population is associated with a small increase in the prevalence of a subclinical hypothyroidism and thyroid autoimmunity [52]. Chronic exposure to excess iodine intake is believed to induce the autoimmunity; it is usually explained with a more pronounced immunogenic character of a highly iodinated thyroglobulin [53]. In especially susceptible individuals, the excess iodine increases intra-thyroid infiltration with Th17 cells and inhibits the development of T regulatory cells (Treg) [54], which is commonly seen in the course of autoimmune thyroiditis.

### 4.3. Selenium

Treatment with L-thyroxine in monotherapy seems to be partly ineffective. The standard substitution protocol with T4 does not reverse the ongoing inflammatory process; moreover, a group of treated individuals declares non-optimal quality of life despite normalization of TSH and thyroid hormone’s levels. Some clinical trials assessing the influence of selected substance addition to standard therapy have occurred. One of such explored agents is selenium—playing an important role in the physiology of the thyroid. Selenium can be perceived as the crucial cofactor of enzymes participating in the production of thyroid hormones—thioredoxin reductase, glutathione peroxidase and type I and II of deiodinases [55]. Having immunomodulatory properties, selenium affects the course of AITD, too. Its level in patients with newly diagnosed HT and GD appeared to be significantly lower compared to healthy individuals. Therefore, a deficiency in selenium could be potentially considered as a risk factor of AITD [55]. The supplementation of selenium in a mice model showed an ability to slow down the ongoing autoimmune process. From a serological point of view, it modified levels of thyroxine and thyroid autoantibodies in peripheral blood and changed the expression of T-cell subsets. Finally, a reduced production of pro-inflammatory cytokines by Th1 cells together with an inhibited differentiation and production of cytokines by Th2 and Th17 cells were noted. Described alterations were responsible for the upregulated differentiation and function of Tregs [56]. In the study conducted by Zhang LY. et al., a combined therapy of L-thyroxine sodium tablets and sodium selenite presented a better clinical efficacy in comparison to a monotherapy with L-thyroxine. Furthermore, a composed option of treatment caused achieved levels of TPOAb, TGAb and TRAb to be significantly lower and accompanying concentration of fT3 and fT4 to be notably higher [57]. According to the last systematic review and meta-analysis of 35 unique randomized clinical trials performed by Huwiler V. et al. B, selenium was proved to lower the level of TSH in patients with no previously substitution therapy with T4 and to lower the concentration of TPOAb and malondialdehyde in patient regardless of L-thyroxine treatment. No significant changes were observed in fT3, T3, fT4, T4, TGAb, interleukin 2 (IL-2) and IL-10 concentrations, nor in thyroid volume [58]. Proper supplementation of the body with selenium determines the correct function of selenoproteins in the organism. What is interesting, autoantibodies to the transporter of selenium (SELENOP) have been detected recently. The mentioned antibodies were shown to be present and to impair both the expression of selenoproteins together with the deiodination of thyroid hormones [59]. These new findings require a further elucidation to define the suitability of the antibodies against the SELENOP for diagnostic and therapeutic purposes. What is extremely interesting, selenium deficiency and elevated iodine in HT are the consequences of autoimmune reactions, leading to thyroid dysfunction through oxidative stress [60]. Adequate selenium intake is especially vital in the areas of iodine deficiency or excess, and in regions of low selenium intake, its supplementation of 50–100 μg/day may be considered a reasonable approach [53]. Simultaneously, other authors are cautious in proposing such an attitude, suggesting that the clinical relevance of TSH normalization and TPOAb reduction is unclear and the long-term safety of selenium supplementation is yet to be established [14]. Of note, the overdosage of the described element may lead to selenosis. This state in individuals may manifest as, for example, hair loss or nails changes.

Nonetheless, new and well-planned clinical trials assessing the influence of selenium on the immune system might help to verify the role of this nutrient as a possible biomarker of autoimmunity in the course of HT.

### 4.4. Vitamin D

Many immunocompetent cells with the example of T and B lymphocytes, dendritic cells, monocytes and macrophages, express the vitamin D-activating enzyme (CYP27B1) and the vitamin D receptor (VDR) [61]. The active form of vitamin D demonstrates the ability to exert direct effects on antigen presenting cells with their representatives [62]. In the general scope, Vitamin D modulates both the innate and adaptive immune system response. It increases the production of anti-inflammatory cytokines (e.g., IL-10) and T-cell inhibitory molecules (programmed death-1, PD-1), and inhibits the secretion of pro-inflammatory cytokines IL-12, IL-23, TNF-α and IFN-γ, which are involved in the differentiation of T helper cells: Th1 and Th17 [62,63]. According to available literature, vitamin D supplementation may significantly lower TSH and total cholesterol levels, as well as reduce fat mass percentage, increasing irisin levels and fat-free mass percentage in women with subclinical hypothyroidism [64]. According to systematic review and meta-analysis of 12 studies, performed by a group of Tang et al., vitamin D supplementation caused the reduction of anti-TPO and anti-TG antibodies in patients with HT. What is more, a decrease in TSH level and increase in both fT3 and fT4 levels were also observed [65]. Calcitriol, which is the active form of vitamin D, seems to have more pronounced influence than D_2_ or D_3_ on the decrease in the titer of TPOAb and increase in fT4 and fT3 levels in patients with HT compared to controls [65]. From a clinical point of view, the supplementary dose of vitamin D should be selected carefully, since the overdosage of this nutrient may lead to hypercalcemia and related adverse effects.

### 4.5. Gut Microbiome and Its Derivatives

Human intestinal microbiota consists of billions of microorganisms, such as bacteria, viruses, fungi and protozoa. Four generic bacteria predominate in the healthy intestinal flora, namely *Firmicutes*, *Bacteroidetes*, *Actinobacteria* and *Proteobacteria* [66]. These subpopulations of microorganisms regulate the function of gut-associated lymphoid tissue (GALT) function and exhibit a close relationship between their representants and a followed diet [67]. GALT remains in close cooperation with gut microbes. This is essential for maintaining proper absorption of nutrients and protection of the mucosa against harmful pathogens. Due to alterations in gut microbiota, intestinal barrier integrity may be weakened or lost. In consequence, intestinal permeability becomes greater, leading to the activation of the immune system, for example via molecular mimicry [68]. GALT effector cells are finally activated and proinflammatory factors produced at that time cause subclinical inflammation. Primarily, the whole process develops in situ only; however, it may spread and transform into persistent generalized inflammation [69]. Fiber is known to be a crucial ingredient necessary for a proper development and metabolism of gut bacteria [70]. The gut microbiome conducts the fermentation of indigestible food components into absorbable metabolites. It is also responsible for the synthesis of essential vitamins and the removal of toxic compounds [71]. Short-chain fatty acids (SCFAs), which are the products of microbial fermentation, act as substrates for enterocytes. Therefore, they behave as modulating factors for immune cells. SCFAs constitutes a direct linkage between gut microbiome and immune system due to their capability for inducting regulatory T-cells (Treg) [72] and regulating Treg’s and Th17 balance [73]. Thus, any state of dysbiosis of the gut microbiome may induce increased intestinal permeability, with a subsequent disruption of SCFAs production, followed by the dysregulation of the immune system. Individuals with a diagnosed HT present with gut dysbiosis. A significant increase in *Bacteroides* species with a concomitant decrease in *Bifidobacterium* among intestinal microbiota have been found in samples of patients [74]. The above-mentioned state is known to be connected with increased gut permeability [75]. Due to the current data, zonulin seems to be a great marker of leaky gut syndrome. Zonulin is a protein, the potential modulator of intestinal tight junctions [76,77]. Elevated levels of this marker were observed in chronic thyroiditis [74,78,79]. What is more, zonulin has been found to correlate with TPOAb levels [80]. With a reference to the above reports, zonulin seems to be a possible biomarker supporting diagnosis and monitoring of treatment in HT patients. However, further analyses are required, mainly due to the fact that zonulin was noticed to be elevated in other various disorders as well, including celiac disease [81], inflammatory bowel disease [77], inflammatory arthritis [82] and hypertension [83]. Larazotide acetate (Zonulin Inhibitor AT-1001), which is an 8-mer peptide and tight junctions (TJ) regulator [84], promotes the correct status of tight junction in epithelial cells [85]. Thus, larazotide acetate appears as a therapeutic option for diseases related to increased intestinal permeability [86]. Once again, further conclusive research is needed concerning such an attempt in the course of thyroid pathologies. As gut microbiota have an undeniable impact on intestinal mucosa integrity and immune system function, there have been attempts in progress to modify a microbiome in several autoimmune diseases, including HT. The effect of prebiotics, probiotics and synbiotics on thyroid function was verified in the meta-analysis of eight controlled trials by the group of Qinxi et al. Researchers did not indicate the intervention on TSH, fT4 and fT3 levels as statistically significant. Yet, modest reduction of TRAb levels in patients with Graves’ disease was noticed [87]. However, this analysis was not dedicated to patients with chronic thyroiditis and levels of TPOAb and TgAb were not assessed. In the subsequent meta-analysis of two randomized controlled trials regarding the influence of probiotics, prebiotics and synbiotics in patients with hypothyroidism, a decrease in TSH (statistically nonsignificant) and change in the levels fT4 (statistically significant) were observed. The outcome of analysis suggests that routine administration of probiotics, prebiotics or synbiotics may result in a very limited benefit among patients with primary hypothyroidism [88]. Treatment with probiotics or SCFAs is not commonly used in autoimmune diseases, whereas there are some data suggesting the positive effect on such a treatment with *Lactobacillus casei* in patients with rheumatoid arthritis [89]. What is more, results of a different study indicated that human-origin probiotics increase the production of SCFAs via modulation of mice and human gut microbiome [90]. It may suggest that this specific kind of therapy may improve gut microbiome status and indirectly affect the immune system. Fecal microbiota transplantation (FMT) is proved to be useful not only in intestinal disorders, but also in metabolic syndrome or in selected immunological diseases [91]. Some interesting surveys concerning the impact of FMT on chronic inflammation in the thyroid and the studies on treatment with probiotics are ongoing [92,93]—we are curious about the results, because they may practically change the way of treatment.

### 4.6. Magnesium

Magnesium is the fourth most common mineral in the human body. It is mainly found within the cell; therefore, it is difficult to assess the magnesium storage in the organism. Magnesium is a cofactor of more than 300 enzymatic systems that regulate diverse biochemical reactions in cells [94]. Deficiency in this mineral in particular individuals has been linked to, e.g., atherosclerosis, dyslipidemia, carbohydrate disorders with type 2 diabetes, myocardial infarction, hypertension, kidney stones, premenstrual syndrome, pre-eclampsia and psychiatric disorders. Furthermore, neurological symptoms are more severe in magnesium-deficient patients [94]. Low serum magnesium levels are also known to be connected with chronic thyroiditis. According to Wang et al., the analysis of the risk of TGAb positivity, HT and hypothyroidism may be higher in patients with lower serum magnesium concentrations [95]. On the other hand, Erdal et al. showed that serum magnesium levels may be significantly higher in the investigated group with subclinical hypothyroidism in the iodine-rich region of Ankara (Turkey) in comparison to controls [96].

Scientific data concerning magnesium deficiency in HT patients are very limited, whereas it provides a field for interesting trials.

### 4.7. Iron

Thyroid peroxidase (TPO) is a transmembrane protein with enzymatic activity, absolutely necessary for thyroid hormone synthesis [97]. TPO becomes active only after binding heme molecules, so its activity depends strictly on the iron status of the individual. AITD patients are frequently iron-deficient due to often co-occurring pathologies, such as autoimmune gastritis or celiac disease. The estimated global prevalence of anemia due to any cause is approximately 24.8%, with higher percentages in children and pregnant women [98]. Iron-deficiency anemia remains the dominant cause of this state [99]. It is followed by the impaired B-cell proliferation [100], altered T-lymphocyte function and weakened adaptive antibody response. These immune defects appear to be fully ameliorated with iron supplementation [101]. In two-thirds of women with persistent symptoms of hypothyroidism, despite appropriate L-thyroxine therapy, lower serum iron concentration is observed; the restoration of serum ferritin above 100 µg/l tends to improve the symptoms [102]. On the other hand, iron overload also may be harmful for immune system function [103]. Coexistence of hypothyroidism and iron deficiency anemia is well known; however, the exact relationship between serum iron status markers and thyroid dysfunction was not indicated [104]. It is important to emphasize that vitamin D supplementation along with thyroid hormones usage in patients with subclinical hypothyroidism improves iron status [102].

### 4.8. Metformin

Metformin, one of the biguanides, is a widely applied antiglycemic drug, particularly in type 2 diabetes. It increases hepatic adenosine monophosphate-activated protein kinase activity, reduces gluconeogenesis and lipogenesis in the liver and simultaneously increases insulin-mediated uptake of glucose in muscles [105]. Metformin therapy in diabetes or in polycystic ovary syndrome has resulted in modest reduction of TSH levels in individuals concomitantly affected by the thyroid disorder [106]. Immunomodulatory effect of metformin is becoming better known nowadays. Regulation the balance of lymphocytes Th17 and Treg, autoantibodies production, macrophage polarization, cytokine synthesis and neutrophil extracellular trap release are only a few ways of action of metformin [107]. The group of Jia et al. explored the therapeutic effect of metformin by assessing the underlying mechanism of immunomodulatory properties of the drug. In a mice model, metformin has reduced TgAb levels and lymphocyte infiltration in thyroid tissue. Furthermore, metformin also significantly suppressed the number and function of Th17 cells and macrophages (M1) polarization in HT mice. The use of this agent altered the intestinal flora of HT mice [108]. Metformin significantly reduces TPOAb and TgAb levels in patients with HT and subclinical hypothyroidism, leading to the reduction in TSH. TPOAb level’s decrease was more pronounced than TgAb reduction [109]. According to the newest data, the beneficial effect of metformin on thyrotropin levels in women with autoimmune thyroiditis may by enhanced with exogenous vitamin D supplementation [110]. Despite the probable beneficial action of metformin exerted on the course of AITD, some contraindications or adverse effects of the drug need to be mentioned. Metformin may increase the risk of lactic acidosis, so starting the treatment with this agent needs to be preceded by careful analysis of renal, liver or heart function of the patient.

### 4.9. Myo-Inositol

Apart from the biochemical improvement, a significant amelioration has been observed in the perception of the symptoms associated with subclinical hypothyroidism as well [111]. Due to the lack of optimization of well-being of the treated individuals, new ways of treatment or supplementation are being investigated. Myo-inositol (Myo) is the first isoform of inositol that has been described, constituting over 99% of the intracellular inositols [112]. It is the precursor for the synthesis of phosphoinositides, which takes part in the phosphatidylinositol signal transduction pathway [112]. Phospholipids, of which inositols are essential elements, constitute cell membranes’ elements. Myo is mainly derived from dietary intake, whereas its endogenous production is also possible [113]. It is involved in different physiological processes, such as neuronal transmission, intracellular effects of insulin or calcium homeostasis [114]. At the thyroid level, it is necessary for appropriate response to TSH; it also regulates indirectly T, B, and Tregs lymphocytes’ behavior, thus exerting an immunomodulatory effect. Myo regulates thyroid hormones’ biosynthesis by the formation of hydrogen peroxide (H_2_O_2_), which is essential for iodine organification in thyrocytes [113]. In addition to the above-mentioned ideas, deficiency of Myo contributes to development of hypothyroidism. The current data concerning favorable impact of Myo supplementation on subclinical hypothyroidism and autoimmune thyroiditis were analyzed by the group of Paparo et al. [113] with a special attention given to the immunomodulatory properties of Myo, such as the reduction concerning levels of thyroid antibodies and pro-inflammatory chemokines (e.g., CXCL10). The influence of Myo has been assessed mainly with selenium co-implementation. In a double-blind randomized controlled trial, the beneficial effects have been obtained due to the selenomethionine treatment on patients affected by autoimmune thyroiditis. The total result was improved by cotreatment with Myo-Inositol [115]. TSH levels have significantly decreased in patients with subclinical hypothyroidism (regardless of the presence of autoimmune thyroiditis), after treatment with Myo and Selenium (Myo + Se), with reduction of antithyroid autoantibodies if they were previously observed. Further analyses suggested the probable reduction of progression of hypothyroidism in subjects with AITD due to therapy of Se and Myo [116]. Further research is needed to assess the exact effect of Myo in thyroid pathologies to justify its potential commonness of administration in future.

### 4.10. Other Nutraceuticals

There is no single proper definition for nutraceuticals, whereas commonly they are described in the literature as “a food (or part of a food) that provides medical or health benefits, including the prevention and/or treatment of a disease” [117]. Some of them belong to the rediscovered “substances used more frequently in the past”. The most common nutraceuticals, which have not been mentioned above, besides myo-inositol, are l-carnitine, melatonin are resveratrol [114]. According to the review performed by Benvenga, the introduction of treatment with l-carnitine may be beneficial in hyperthyroid patients, especially symptomatic individuals, who need to take lower doses of antithyroid drugs (2 g/day). When it comes to melatonin and resveratrol, current data are insufficient to determine conclusively if these substances can be undeniably described as beneficial in thyroid disease. The positive influence of *Nigella sativa* as an immunomodulator (with its ingredient—thymoquinone) on thyroid function also should be shortly mention here [118].

### 4.11. Diet and Thyroiditis

As mentioned above, diet is a significant modifiable environmental factor potentially involved in the onset and the course of autoimmune thyroiditis. Of note, food intake directly affects the gut microbiome; these disorders are mainly caused by processed food [70]. Additionally, other common biological factors, e.g., lifestyle or a general hormonal status, are related to the presentation of the disease. According to the data analyzed by Osowiecka and Myszkowska-Ryciak, the cessation of energy due to deficient diet can improve the results of TPOAb, TSH and fT4 levels. Similar results were observed after the elimination of ingredients such as gluten, lactose or goitrogens. A beneficial effect was also noted after the introduction of *Nigella sativa* to the diet with no such improvements observed in patients on the iodine-restricted diet [119]. Interestingly, a paleolithic diet seemed to have positive effect on clinical improvement, as it is a natural source of nutrients similar to supplements, improving the function of thyroid gland in HT and Graves’ disease natural history [120]. In turn, meat consumption has been suggested to be associated with increased odds ratio of developing thyroid autoimmunity. Additionally, the Mediterranean diet could play a positive, protective role concerning the involvement of OS in the natural history of AITD [121]. Analyzing the above data, it seems reasonable to include, in everyday diet, products rich in selenium, iron or vitamin D. With a view to the role of oxidative stress on HT development, products rich in antioxidants should be implemented (e.g., fruit, vegetables, herbs and spices, mostly with a high concentration of polyphenols).

### 4.12. Thyroid Gland, Obesity and Lifestyle Intervention

Environmental factors are known to participate in the pathogenesis of chronic thyroiditis. Therefore, modifications of lifestyle should be considered as an inseparable part of treatment in patients with HT. In the review performed by Matlock et al., hormonal treatment and lifestyle interventions have been explored and compared to each other as options for the reduction of symptoms due to subclinical hypothyroidism in women. In the discussed survey, both approaches appeared to be effective. Lifestyle modifications taken under consideration included the intensification of physical activity, smoking cessation, improved sleep hygiene and a diet with adequate supply of iodine and selenium [122]. Obesity constitutes an undeniable risk factor for numerous disorders, including cardiometabolic disturbances [123], dysregulation of iron homeostasis [124], different types of cancer [125] and many more. Excess body weight is also a predisposing state for hypothyroidism in patients with chronic autoimmune thyroiditis [126]. In accordance to available medical knowledge, obesity appears to be a major environmental factor contributing to the onset and progression of autoimmune diseases [127]. Due to the research performed by Walczak et al., selected adipokines (e.g., leptin, chemerin and the leptin/adiponectin ratio) have not been related to the presence of autoimmune thyroiditis [128]. Conversely, in other investigations, leptin was found to be involved in the development and progression of many autoimmune conditions [129,130]. Leptin belongs to the family of adipokines (a kind of cytokine produced by adipocytes) with established influence not only on endocrine homeostasis, but also on immune balance of the host. Its elevated circulating level in obese individuals seems to trigger the low-grade inflammatory milieu, developing under the mediation of T helper 1 cells [130]. In the study of Wang et al., the antagonist of leptin’s receptor in mice model was studied. The clinical effect of this agent was responsible for the altered differentiation of Treg/Th17 cells by inhibiting the leptin signaling pathway. Thus, it was concluded to be a potential approach alleviating thyroid injury [131]. The authors suggested that the leptin signaling pathway may be considered as a novel approach to treat chronic thyroiditis. Meanwhile, before the formulation of new therapeutic guidelines concerning leptin, the introduction of a diet notably decreasing its concentration in the organism seems to be a rational alternative.

### 4.13. Thyroidectomy and Thyroid Gland Transplantation

As some patients do not achieve the optimization of results at the time of adequate therapy with L-thyroxine, a suspicion might be raised that the ongoing immune process constitutes a cause of persistence of symptoms. Guldvog et al. have conducted research comparing thyroidectomy versus medical management for euthyroid patients with HT and persisting symptoms. Researchers demonstrated the improvement in the health-related quality of life and reduction of fatigue in the group of patients after thyroidectomy; positive effects were explained with the elimination of serum anti-TPO antibodies [132]. Similar positive observations were noticed in the survey conducted by Hoff et al. Total thyroidectomy performed in patients with HT and an achieved euthyreosis due to pharmacological treatment was proved to exert a beneficial and long-lasting effect up to 5 years, concerning general health score. Therefore, achievement of normalized laboratory markers concerning thyroid’s function do not have to be related with the disappearance of extrathyroidal symptoms (e.g., fatigue, dry mouth, eyes muscle and joint tenderness and impaired sleep quality) [133]. This invasive and irreversible way of treatment could be potentially considered; however, currently, it is not a recommended approach if there are no other serious indications for surgery, such as thyroid compression or suspicion of thyroid neoplasm.

## 5. Conclusions

HT is an autoimmune disease with multifactorial and still not entirely known etiology. Further investigations concerning pathogenesis of the disease will open the field for an exploration of new possible ways of treatment. A more precise determination of the genetic background according to HT will probably become a milestone in choosing the treatment option for the individual patients. The precise interpretation of environmental factors influencing the onset and the course of thyroiditis will contribute to proposing certain lifestyle changes, enabling the prevention and slowing down the course of HT in the relation to genetic predispositions. L-thyroxine substitution therapy is a well-known and accepted way of treatment of hypothyroidism due to AITD. Recent data suggest this drug to be an agent exerting a notable immunomodulatory effect. However, the implementation of T4 cannot be perceived as an entirely curative approach for each mechanism underlying the pathogenesis of HT. Due to the current knowledge, the introduction of some alternative substances or supplements seems to be worth considering in the treatment of patients with thyroid immunity, especially when nutritional deficiencies are recognized. On the other hand, when a patient’s well-being cannot be optimized with ongoing treatment, the awareness of the co-occurrence related to different autoimmune diseases has to be underlined.

Simultaneously, this immune background of thyroid pathologies was shown to be inseparably related to OS. Therefore, besides the addition of pharmacological cofactors acting against the development of free radicals, oxidative derivatives are explored as potential markers of thyroid disease. The incidence of AITD with chronic autoimmune thyroiditis is rapidly growing worldwide. As a consequence of this observation, new trustworthy biomarkers confirming the diagnosis, indicating the progression and the possible outcome of an illness, are urgently needed.

## Figures and Tables

**Figure 1 ijms-25-04703-f001:**
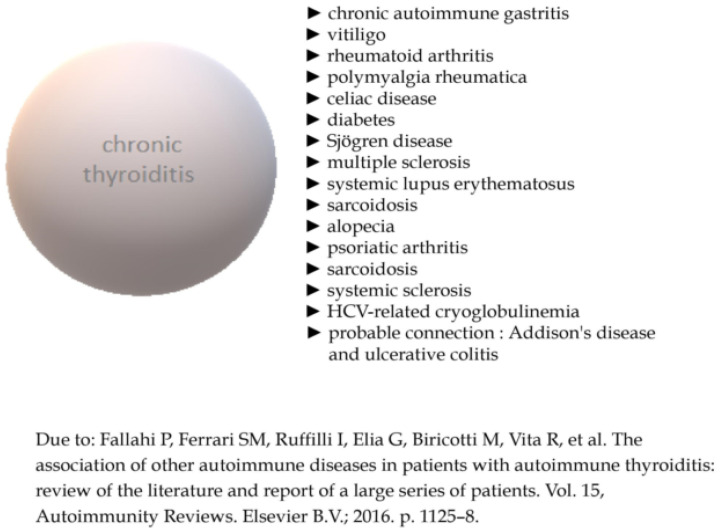
Autoimmune disorders related to HT [15].

**Figure 2 ijms-25-04703-f002:**
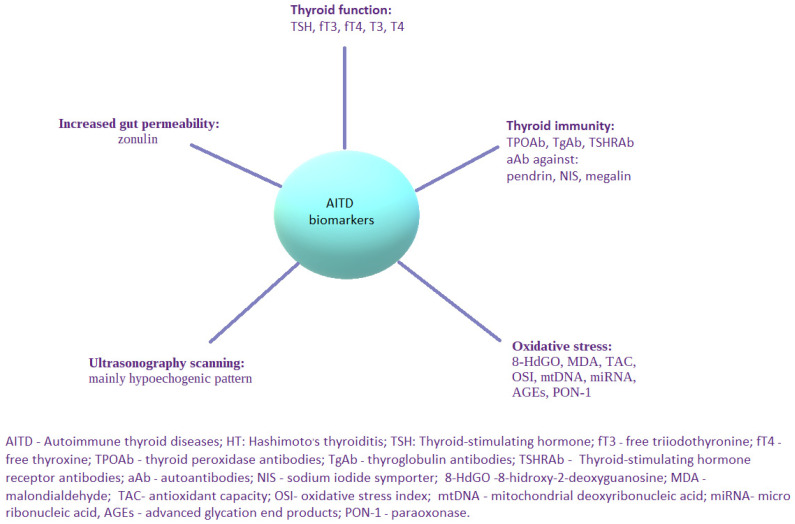
Possible biomarkers applicable in AITD, with a special focus on HT.

## Data Availability

Not applicable.

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
