# Peer review of "Autoimmunity, New Potential Biomarkers and the Thyroid Gland—The Perspective of Hashimoto’s Thyroiditis and Its Treatment"

_ijms, 2024, doi:10.3390/ijms25094703_

Round 1

Reviewer 1 Report

Comments and Suggestions for Authors

REVIEW:  IJMS #2945189 Tywanek et al.

      In this manuscript, Dr. E. Tywanek and co-authors provide a review of thyroiditis focusing on potential biomarkers of underlying pathology, plus recent advancements in the use and/or understanding of natural or nutraceutical interventions in merging treatments. Overall, this manuscript goes beyond the general autoimmune aspects of thyroiditis, delving into multiple non-autoimmune biological activities that could become critical future targets for intervention therapies. A major strength of this review is reflected in its discussion / presentation of individual biomolecules and/or biopathways that are becoming a potential focus of underlying pathological involvement. Potential weaknesses are the lack of discussions concerning possible negative effects within the discussed focuses, as well as the why and how these approaches are or could be superior to current treatment with, e.g., Synthroid, L-thyroxine,  Eutirox, etc.  

     While the manuscript provides a decent overview of the subject information, the details tend to get lost in the overall presentation, in great part due to the organization.  This Reviewer suggests that the manuscript would benefit considerably if slightly rearranged as (a) an in-depth description of thyroiditis, (b) the possible links of diet and general biological factors with disease states, (c) what constitutes specific disease-influencing factors, (d) what are specific molecular entities that have undergone trials and their overall effect on disease pathology, (e) potential future pathways and interventions that will better our understand of thyroiditis….in this case Hashimoto´s thyroiditis, and (f) what can be concluded thus far.

     Although the Authors have provided a fairly well-written manuscript, it is recommended 

Comments on the Quality of English Language

     Although the Authors have provided a fairly well-written manuscript, and one that this reviewer appreciates, there are a number of areas that require minor grammar editing.

Author Response

Thank you for your valuable review that improved the general shape of the manuscript. We tried to follow all of your suggestions:

  1. Additional comments concerning new approaches of the treatment in patients with thyroiditis (together with their potential side effects) were included. → lines: 291, 315 and 441
  2. A detailed description of thyroiditis was added. → lines: 34 and 42
  3. We included information concerning possible links of diet and general biological factors with the course of thyroiditis. → line 495
  4. The information according to the thyroiditis-influencing factors was included. → line 48
  5. Molecular pathways were added, as well. → line 121
  6. We included data on potential future pathways and interventions. → line 564
  7. Conclusions regarding the current state of knowledge in case of treatment patients with thyroiditis have been included, too. → line 564
  8. The grammar editing of the whole manuscript was performed.

Reviewer 2 Report

Comments and Suggestions for Authors

Thanks the Editor to give me the opportunity to revise this article. The manuscript is of great interest in the field of current research. The Authors performed a literature review with the aim of investigating and collecting the current data in the literature regarding the lateste knowledge about the diagnosis and treatment of HT.
The work is well written and adequately structured. The literature review was carried out correctly and examined an adequate number of articles. The conclusions are well written and represent a clear and appropriate conclusion to the review. The figures are well constructed, clear and provides a relevant contribution to the whole article. 

Having said this, I suggest authors to updated the last paragraph "4.13", citing the newest results of the study (PMID: 38011703), regarding the 5 years follow-up of patients treated or not with thyroidectomy for euthyroid HT.

Author Response

Thank you for your valuable comments that improved the manuscript. We tried to follow all the suggestions. 

We included in the main body of the manuscript the results of indicated research (PMID: 38011703). → line 552